# Lasso Analysis of Gait Characteristics and Correlation with Spinopelvic Parameters in Patients with Degenerative Lumbar Scoliosis

**DOI:** 10.3390/jpm13111576

**Published:** 2023-11-03

**Authors:** Chen Guo, Yan Liang, Shuai Xu, Bin Zheng, Haiying Liu

**Affiliations:** Department of Spinal Surgery, Peking University People’s Hospital, Peking University, Beijing 100044, China; fantasy_g@pku.edu.cn (C.G.); liangyan75@pkuph.edu.cn (Y.L.); xushuairmyy@pku.edu.cn (S.X.); zhengbin97@stu.pku.edu.cn (B.Z.)

**Keywords:** degenerative lumbar scoliosis, lumbar stenosis syndrome, Cobb angle, balance, gait

## Abstract

Purpose: This study quantifies the gait characteristics of patients with degenerative lumbar scoliosis (DLS) and patients with simple lumbar spinal stenosis (LSS) by means of a three-dimensional gait analysis system, aiming to determine the image of spinal deformity on gait and the correlation between spinal–pelvic parameters and gait characteristics in patients with DLS to assist clinical work. Methods: From June 2020 to December 2021, a total of 50 subjects were enrolled in this study, of whom 20 patients with DLS served as the case group and 30 middle-aged and elderly patients with LSS were selected as the control group according to the general conditions (sex, age, and BMI) of the case group. Spinal–pelvic parameters were measured by full-length frontal and lateral spine films one week before surgery, and kinematics were recorded on the same day using a gait analysis system. Results: Compared to the control group, DLS patients exhibited significantly reduced velocity and cadence; gait variability and symmetry of both lower limbs were notably better in the LSS group than in the DLS group; joint ROM (range of motion) across multiple dimensions was also lower in the DLS group; and correlation analysis revealed that patients with a larger Cobb angle, T1PA, and higher CSVA tended to walk more slowly, and those with a larger PI, PT, and LL usually had smaller stride lengths. The greater the PI-SS mismatch, the longer the patient stayed in the support phase. Furthermore, a larger Cobb angle correlated with worse coronal hip mobility. Conclusions: DLS patients demonstrate distinctive gait abnormalities and reduced hip mobility compared to LSS patients. Significant correlations between crucial spinopelvic parameters and these gait changes underline their potential influence on gait disturbances in DLS. Our study identifies a Cobb angle cut-off of 16.1 as a key predictor for gait abnormalities. These insights can guide personalized treatment and intervention strategies, ultimately improving the quality of life for DLS patients.

## 1. Introduction

Degenerative lumbar scoliosis (DLS) is a common disease in the elderly, characterized by coronal malformations of the lumbar or thoracolumbar spine due to degeneration in adulthood, affecting 32% to 68% of population over 65 years old [1]. The initial factor for DLS may be the degeneration of the intervertebral disc, which leads to the loss of intervertebral space, ligament laxity, and asymmetric articular processes. These changes result in the inability of functional units to maintain normal [2] alignment of the spine, causing vertebral body tilt and resulting in scoliosis [3].Typical symptoms of DLS include progressive deformity, back pain, and lower extremity pain [4], accompanied by characteristic neurogenic claudication and relief of symptoms after rest [5]. As the disease progresses, spinal deformities can lead to abnormalities in gait.

Surgical treatment for patients with DLS has two main goals: releasing spinal stenosis through decompression fusion and correcting spinal deformity through long-segment fixation. Therefore, detailed physical examination and appropriate imaging studies, such as magnetic resonance imaging (MRI), should be performed on patients with DLS [6]. However, MRI findings often do not match patients’ symptoms. In recent years, spinopelvic parameters, such as pelvic incidence (PI), lumbar lordosis (LL), PI-LL mismatch, pelvic tilt, and sagittal vertical axis (SVA), have been increasingly recognized [7,8]; function scales have also been used to assess patients with lumbar degenerative diseases [9]. However, radiographic tests can only evaluate the static equilibrium state, and scales are susceptible to subjective factors such as patients’ experience and a physicians’ clinical expertise [10].

Diebo et al. have emphasized the importance of the dynamic aspects of alignment when managing spinal deformities [11], and both dynamic aspects of alignment and clinical evaluation are crucial to the management of spinal deformity conditions [12]. Walking is very common dynamic aspects for healthy people, but it is actually a complex casual activity [13]. The gait cycle consists of two phases: the stance phase (60%) and the swing phase (40%). Gait analysis has become widely used in the diagnosis and evaluation of clinical diseases. Quantified gait and balance characteristics can be a useful tool to improve the evaluation and understanding of the biomechanical effects of DLS [14,15]. Recent studies have shown that patients with DLS have increased single support time, center of mass, head sway, and neuromuscular activity [12]. Patients with adolescent idiopathic scoliosis (AIS) have also been reported to have variations in gait speed and length compared to normal, with asymmetric variations [16,17]. In addition to structural changes, patients with DLS are usually associated with lumbar spinal stenosis, leading to symptoms such as intermittent claudication [18]. Both spinal deformities and spinal stenosis coexist and have a corresponding effect on the patient’s gait. Few articles distinguish what factors cause gait changes in DLS patients. Therefore, we used a portable gait analysis device to measure the gait characteristics of a group of ADS and LSS patients, focusing only on the influence of spinal structural deformities on gait (excluding intermittent claudication), and then analyzed the correlation between these gait parameters and spinal–pelvic parameters in DLS patients to assist clinical work. Our study addresses this gap by employing advanced statistical techniques like Lasso and machine learning, providing a more nuanced understanding of the factors affecting gait in DLS patients.

## 2. Materials and Methods

Following approval from the institutional review board committee, we prospectively recruited patients with DLS who presented to our clinic and were deemed surgical candidates.

### 2.1. Participants

This single-center prospective study was conducted from June 2020 to December 2021 following approval from the institutional review board committee. Participants diagnosed with DLS (degenerative lumbar scoliosis) and LSS (lumbar spinal stenosis) who were deemed surgical candidates were recruited from our institution.

Control Group Clarification: LSS patients requiring surgery but without spinal deformity were selected as the control group. Importantly, all LSS patients in this study exhibited symptoms of intermittent claudication, thereby serving as a suitable control to counteract the effect of intermittent claudication observed in the DLS group.

All participants provided informed consent prior to enrollment, and the study was approved by the ethics committee at our institution.

Inclusion criteria are (1) DLS group are patients clinically diagnosed thoracolumbar deformity, with a Cobb angle of ≥15 degrees; (2) patient age  ≥  55 years; (3) surgical intervention is recommended for symptoms; (4) had complete and clear preoperative X-ray of the whole standing spine and thoracolumbar or lumbar MRI; and (5) are able to ambulate without assistance. (6) All patients were receiving conservative treatment at study entry, including general physiotherapy or pain control through analgesics. Exclusion criteria are (1) diagnosis of idiopathic scoliosis, congenital scoliosis or other spinal deformities; (2) patients with lumbar spine fracture, tumor, infection or spondylolisthesis; and (3) those who have undergone thoracolumbar surgery or major lower extremity surgery that affect ambulation.

### 2.2. Parameters Measurement

The standard radiographs were undertaken using a long-cassette standing radiographs of the spine generating anterior/posterior as well as lateral views, before-surgery, including spinal parameters and pelvic parameters.

T1PA: the sum of the pelvic tilt and the angle between the line from the hip axis to the center of Tl and the plumb line from the hip axis;

SVA: offset horizontally from a plumb line dropped from C7 to the posterosuperior corner of the sacrum;

CSVA: offset horizontally from a plumb line dropped from C7 midline to the center of the sacrum;

TK: the sagittal angle between the superior endplate of T5 and the inferior endplate of T12, which was a positive value in kyphosis patients;

TLK: the sagittal angle between the superior endplate of T10 and the inferior endplate of L2;

LL: the sagittal angle between the superior endplate of L1 and the inferior endplate of S1, which was a positive value in lordosis patients.

PT: the angle between the line from the center of the femoral head to the upper endplate of S1 and the plumb line;

PI: Draw a vertical line through the midpoint of the upper endplate of S1. The angle between the vertical line and the midpoint of the upper endplate of S1 and the femoral head center.

SS: the angle between the upper endplate of S1 and the horizontal line;

These radiographic parameters were then measured on the PACS client software (Easy Vision IDS5, version 11.4; Philips, Hamburg, Germany) by 2 investigators independently.

### 2.3. Gait Measurement

The test subjects were all fitted with a total of 7 external reflective sensors (Figure 1). One at the waist and one for each of three lower limb joints at one side, we attach to the bony positions to avoid movement during the test. The patients are recommended to walk for at least 2 min to collect enough gait cycles to analyze the variability and asymmetry of the gait. The patient needs to walk normally on a straight horizontal path which is straight and flat with no obstructions. The length of the path is 10–50 m and it is optimal to be longer than 25 m. The patient can turn around and walk back when getting to the end of the path and do this for several times. The total walking distance reaches 50 m. To account for intrasubject variability, each patient performed the test 2 times and the average was collected.

The figure demonstrates the positioning of the external reflective sensors used for gait analysis. Reflective sensors were placed at the waist and three lower limb joints on one side of the body. These sensors allowed for precise measurement of kinematic data during walking, enabling the assessment of gait characteristics and mobility parameters in patients with degenerative lumbar scoliosis (DLS) and lumbar spinal stenosis (LSS).

### 2.4. Statistical Analysis

Descriptive statistics were calculated for all variables, and the Shapiro–Wilk test was used to test for normal distribution. Continuous variables were expressed as means ± standard deviation (SD). An independent sample *t*-test was used to compare parameters between DLS patients and LSS patients. For the analysis of the relationship among spinopelvic radiographic parameters and gait characters in patients with DLS, lasso regression and Pearson correlation analysis were used. The lasso regression model was used to identify the most important radiographic parameters associated with gait characters. Pearson correlation analysis was used to determine the correlation between spinopelvic radiographic parameters and gait characters, and the strength of the correlation. To investigate the relationship among spinopelvic radiographic parameters, range of motion variables, and gait variables in patients with DLS, mediation analysis was conducted to determine if the effect of spinal variables on gait variables is mediated by ROM variables. ROC analysis was added to determine cut-off values for key spinopelvic parameters like the Cobb angle, assessing their diagnostic performance through AUC, sensitivity, and specificity.

All statistical analyses were conducted using R version 4.1.1 (R Foundation for Statistical Computing, Vienna, Austria). A two-sided *p*-value < 0.05 was considered statistically significant.

## 3. Results

A total of 20 patients in the DLS group and 30 cases in the LSS group were included in this study. There was no statistically significant difference in gender, age, and BMI between the two groups (*p* > 0.05) (Table 1).

### 3.1. Parameters Measurement between DLS and LSS Patients

Apart from Cobb angle, when comparing parameters between DLS and LSS patients, there was a significant difference in the CSVA, T1PA, LL, PT, SS, and PI-LL. The CSVA was significantly larger in the DLS group (27.73 ± 5.58) than in the LSS group (3.52 ± 5.59). The T1PA was twice as large in the DLS group (32.29 ± 9.26) than in the LSS group (16.55 ± 7.61). The LL was twice as small in the DLS group (21.76 ± 5.09) than in the LSS group (44 ± 4.02). The PT and SS were also significantly different, which led to a significant difference in the PI-LL between the DLS group (32.2 ± 4.77) and the LSS group (12.71 ± 1.13) (Table 2).

### 3.2. Kinematic Gait Variables between DLS and LSS Patients

During walking, the DLS group exhibited significantly reduced hip ROM in the sagittal (DLS: 26.17° ± 6.18°, LSS: 32.83° ± 4.40°), coronal (DLS: 16.32° ± 5.78°, LSS: 20.48° ± 6.75°), and axial planes (DLS: 6.23° ± 3.48°, LSS: 10.58° ± 4.68°) compared to the LSS group. On the other hand, the sagittal ROM of the knee was significantly larger in patients with LSS (55.91 ± 5.14) than in DLS patients (46.91 ± 9.35). Pelvic anteversion was also significantly increased in patients with DLS (6.10 ± 2.44) compared to LSS patients (3 ± 1.82). No differences in ROM were found for the lumbar spine, knees, or ankles, except for a few indicators (Table 3).

### 3.3. Gait Ability and Pattern Analysis between DLS and LSS Patients

Preoperative gait ability was significantly worse in patients with DLS compared to the LSS group. During gait, both groups demonstrated similar stride length (LSS: 0.85 ± 0.29 m, DLS: 0.70 ± 0.26 m), total stance phase (LSS: 58.30%, DLS: 59.25%), total swing phase (LSS: 41.70%, DLS: 40.75%), and double stance phase (LSS: 14.22%, DLS: 14.85%). However, there was a significant difference in cadence between the DLS (87.05 ± 2.61) and LSS (102.95 ± 13.98) groups. Additionally, the velocity of the DLS group were significantly lower than those of the LSS group, indicating the impact of spinal deformity on gait.

We observed different gait parameters in the comparison of bilateral lower extremities between the two groups. A similar gait variability was found in the LSS (3.45%) group, but a higher gait variability was observed in the DLS group (9.53%). Furthermore, for gait symmetry, there was little difference in bilateral lower extremities (4%) in the LSS group compared to the DLS group (8%) (Table 4).

### 3.4. Correlation of Spinopelvic Parameters with Gait Parameters in Patients with DLS

#### 3.4.1. Velocity

Lasso regression is a biased estimation regression used to solve covariance problems: parameter λ = 0.364 (Figure 2), the variable intercept term, Cobb, T1PA, and PT are retained (Figure 3).

The figure visualizes the process of selecting the optimal λ value using cross-validation. The vertical axis represents the model mean squared error, while the horizontal axis represents the logarithmic values of λ. The plot provides a visual representation of the relationship between λ and the model’s performance, aiding in the determination of the most suitable regularization parameter for the analysis.

The figure illustrates the changes in model coefficients as the logarithmic values of λ vary.

The Pearson analysis shows the following:The Cobb angle was correlated with velocity (r = −0.612, *p* = 0.015)The CSVA was correlated with velocity (r = −0.522, *p* = 0.046)The T1PA was correlated with velocity (r = −0.636, *p* = 0.011)The PT was correlated with velocity (r = −0.563, *p* = 0.029)The sagittal ROM of the hip as correlated with velocity (r = 0.613, *p* = 0.015)

The multiple linear regression analysis shows the following:Velocity = y = 1.138 − 0.005 × Cobb − 0.005 × T1PA − 0.007 × PT (F = 8.797, *p* = 0.003)

This was significantly correlated due to hip sagittal ROM and Cobb PT. They were examined for parallel mediating effects and no significant mediators were found.

#### 3.4.2. Cadence

Lasso regression is a biased estimation regression used to solve covariance problems: parameter λ = 0.741, variable intercept terms, Cobb, SS, sagittal ROM of the hip, coronal ROM of the waist are retained.

The Pearson analysis shows the following:The SS was correlated with cadences (r = 0.468, *p* = 0.049), the mediation analysis revealed that waist coronal ROM was fully media)The sagittal ROM of the hip was correlated with cadence (r = −0.498, *p* = 0.049)The coronal ROM of the waist was correlated with cadence (r = −0.666, *p* = 0.007)

The multiple linear regression analysis show the following:Cadence = 145.88 + 0.48 × SS − 1.251 × sagittal ROM of the hip − 3.041 × The coronal ROM of the waist (R^2^ = 0.714, *p* = 0.025)

The Cobb angle was correlated with waist coronal ROM (r = −0.499, *p* = 0.058); the mediation analysis revealed that waist coronal ROM fully mediated the relationship between the Cobb angle and cadence.

#### 3.4.3. Stride Length

Lasso regression is a biased estimation regression used to solve covariance problems: parameter λ = 0.364, variable intercept terms, PT, LL, sagittal ROM of the hip, axial ROM of the hip, sagittal ROM of the knee, sagittal ROM of the waist are retained.

The Pearson analysis shows the following:The PT was correlated with stride length (r = −0.601, *p* = 0.018)The LL was correlated with stride length (r = −0.564, *p* = 0.029)The sagittal ROM of the hip was correlated with stride length (r = 0.659, *p* = 0.008)The axial ROM of the hip was correlated with stride length (r = 0.562, *p* = 0.029)

The multiple linear regression analysis shows the following:stride length = 0.569 − 0.017 × PT + 0.016 × sagittal ROM of the hip + 0.01 × axial ROM of the hip (F = 13.183, *p* = 0.001)

This is significantly correlated due to waist sagittal ROM and LL. They were examined for parallel mediating effects and no significant mediators were found.

#### 3.4.4. Total Stance Phase

Lasso regression is a biased estimation regression used to solve covariance problems: parameter λ = 2, variable intercept terms, LL, PI-LL, sagittal ROM of the waist are retained.

The Pearson analysis shows the following:The LL was correlated with total stance phase (r = 0.612, *p* = 0.015)The PI-LL was correlated with total stance phase (r = −0.488, *p* = 0.045)The sagittal ROM of the hip was correlated with total stance phase (r = 0.519, *p* = 0.047)total stance phase = 60.286 − 0.136 × PI-LL + 0.488 × sagittal ROM of the waist

#### 3.4.5. Gait Variability and Symmetry

The Cobb angle was correlated with gait symmetry (r = −0.517, *p* = 0.048) during walking.

The T1PA was correlated with gait variability (r = 0.542, *p* = 0.037).

### 3.5. ROC Analysis of Spinopelvic Parameters Affecting Gait

A receiver operating characteristic (ROC) analysis was performed to identify the predictive value of the Cobb angle for gait abnormalities, specifically for walking speeds lower than 0.8 m/s, in DLS patients [19].

The area under the curve (AUC) for the Cobb angle was 0.751, suggesting a good ability to discriminate between patients with and without gait abnormalities. Importantly, the optimal cut-off value for the Cobb angle was identified as 16.1. This cut-off value demonstrated a sensitivity of 0.692 and a specificity of 0.800, indicating its robustness as a diagnostic criterion. The cut-off value of 16.1 for the Cobb angle serves as a critical threshold for clinicians. Patients with a Cobb angle greater than this value are at a higher risk for gait abnormalities and may require more aggressive intervention strategies. This cut-off can be instrumental in early identification and targeted treatment, thereby potentially improving the quality of life for DLS patients (Table 5).

## 4. Discussion

Degenerative lumbar scoliosis (DLS) is a highly prevalent disorder in middle-aged and elderly populations [4], and its management is still inconclusive due to its complexity [11]. Diebo et al. proposed that, for the proper assessment and treatment of DLS, identifying dynamic parameters in addition to static ones and determining their correlation with function is crucial. Spinal deformities caused by DLS can negatively impact patients’ quality of life, but there is still a lack of quantifiable assessment techniques for patient function in clinical practice [1]. Our study is innovative in that it is one of the first to compare gait characteristics and hip mobility between DLS and LSS patients in China, while also examining the influence of key spinopelvic parameters. Unlike previous studies that have primarily focused on sagittal parameters, our research provides a comprehensive analysis of both, offering a more holistic view of the condition.

Gait-motion analysis can provide clinicians with a wealth of data and can help in identifying functional deficits in DLS patients [1]. While some studies have examined the gait of adults with scoliosis, most of them have set the control group as normal individuals [20], which ignores the symptoms of claudication caused by spinal degeneration. To address this issue, this study used lumbar spinal stenosis (LSS) as the control group. The outcomes of this study showed that the deformities caused by DLS affect not only static balance but also dynamic function.

### 4.1. Spinal Parameters

Previous study shows sagittal spine alignment parameters, rather than coronal Cobb angle, have a greater impact on disability and quality of life, as confirmed by several studies [21,22].

The present study’s findings on sagittal imbalance are supported by previous studies, which add credibility to the next step gait study. The study found that patients with degenerative lumbar scoliosis (DLS) had only half the lumbar lordosis (LL) of the control group. This reduction in LL is a common finding in patients with adult spinal deformity (ASD), which results in forward trunk movement, leading to an increased sagittal vertical axis (CSVA) [23,24]. The increase in CSVA and T1 pelvic angle (T1PA) indicates sagittal imbalance in DLS patients. Sagittal imbalance is a condition in which the body’s weight-bearing axis is shifted forward, resulting in abnormal posture and gait. To compensate for this imbalance, patients tend to tilt their pelvis back to reposition the center of gravity on their feet while standing, as shown in previous studies [23]. In line with this compensation mechanism, the present study found that pelvic tilt (PT) was significantly increased in the DLS group compared to the control group.

The most clinically relevant and strongly correlated radiological parameters with HRQOL were PT, CSVA, and PI-LL, as supported by previous studies on adult spinal deformities [25]. In this study, PT and PI-LL in the DLS group were twice as high as in the control group, and CSVA in the DLS group was several times higher than in the control group. These results are consistent with previous studies on adult spinal deformities.

When it comes to dynamic function, the results of the study show that the preoperative gait ability of the DLS group was worse compared to the LSS group.

### 4.2. Velocity

The slower gait velocity (0.65 ± 0.15 m/s vs. 0.77 ± 0.23 m/s) observed in the DLS group compared to the LSS group indicates the impact of spinal deformity on gait, as suggested by the results of the Lasso regression analysis (Cobb r = −0.612, *p* = 0.015; CSVA r = −0.522, *p* = 0.046; T1PA r = −0.636, *p* = 0.011), which showed that both coronal and sagittal positions have an effect on velocity. indicating that coronal and sagittal positions together have an effect on velocity. Sagittal imbalance, as studied by Semaan et al. on ASD, has been found to be negatively correlated with velocity (CSVA PT), further supporting the association between gait velocity and sagittal imbalance [26]. Loss of lumbar lordosis due to disc degeneration, as often seen in the elderly, can lead to sagittal imbalance such as CSVA PT, which can subsequently affect gait velocity. Therefore, restoring sagittal balance through surgery is necessary for patients with ASD.

While Miura et al. showed that gait balance in the spine with sagittal imbalance deteriorated significantly with trunk advancement, compared with coronal changes, that did not affect spine balance and gait [27]; however, the results of this study suggest that coronal imbalance also affects gait velocity. The cobb angle was found to significantly affect velocity, which is consistent with the study of Mahaudens et al. on AIS [28]. In their study, they found that the quadriceps, erector spinae, gluteus maximus, and semitendinosus muscles showed a bilateral increase in electrical activity (EMG) duration (*p* < 0.001) (46% of stride length in scoliosis patients and 35% of stride length in healthy subjects) [28]. The gluteus maximus plays a crucial role in hip extension, generating forward momentum during the swing phase of gait [29]. The quadriceps, on the other hand, play a key role in the stance phase of gait and can influence gait velocity through the speed and efficiency of their contraction [30]. This may explain why coronal surface deformity can affect gait speed.

### 4.3. Cadence

Gait cadence, which refers to the number of steps taken per minute, is an important aspect of gait that has been found to be strongly correlated with gait velocity [31]. Despite velocity, the study also found that the DLS group had a lower cadence (87.05 ± 2.61 apm vs. 102.95 ± 13.98 apm) than the LSS group just. Muscle abnormalities caused by the coronal plane as shown above can also affect gait cadence, as the quadriceps muscle group [32] and the hip flexor muscles [33], including the iliopsoas and rectus femoris, play a role in regulating gait cadence. In addition, the study found that sagittal imbalance, as measured by the SS was also correlated with cadence (r = 0.468, *p* = 0.049). The compensatory changes caused by the pelvis rotating posteriorly around the femoral head can result in the sacrum tilting posteriorly, which in turn can affect gait cadence. The SVA, which is a measure of sagittal balance, has also been associated with a decrease in gait cadence [31]. DLS patients had a significantly slower gait cadence than middle-aged and older healthy people, probably related to the general imbalance of ASD patients who could not adjust their body balance in time to stand or walk after the gait cadence was accelerated.

### 4.4. Gait Variability and Symmetry

Gait variability and asymmetry were significantly higher in the DLS group than in the LSS group, This also explains to some extent the poorer gait of DLS, because a large variability or asymmetry means extra muscles do work [27].

Gait variability refers to the fluctuation of gait parameters between consecutive steps during walking. We found the T1PA was correlated with gait variability (r = 0.542, *p* = 0.037), Similar studies have confirmed its relationship with sagittal imbalance, in a study by Mickle et al., a higher SVA was found to be associated with increased stride time variability in older adults [34]. Another study by Dodelin et al. found that greater pelvic tilt was associated with increased gait variability in individuals with low back pain [35].

On the other hand, gait symmetry, which is the consistency of step length and duration between the left and right sides of the body during walking was found to be correlated with the Cobb angle in the study (r = −0.517, *p* = 0.048). Although no other studies have been conducted on gait symmetry in patients with DLS, previous research on patients with adolescent idiopathic scoliosis (AIS) provides some insights. A study by Yang et al. found that the scoliosis group had an asymmetrical gait compared to the control group [17], and Zhang et al. also found that scoliosis patients had significantly decreased gait symmetry compared to healthy controls. The degree of spinal curvature was significantly correlated with gait symmetry, with greater curvature leading to decreased symmetry [17]. These findings suggest that spinal deformities may affect overall postural control strategies, resulting in asymmetrical gait patterns.

However, the study found that, during gait, both groups demonstrated similar stride length and total stance phase.

### 4.5. Stride Length

Stride length is defined as the distance between the heel strike of one foot and the subsequent heel strike of the same foot during a gait cycle. It is considered an important parameter for assessing gait patterns and is associated with several clinical outcomes, including balance, fall risk and overall mobility. This finding may be surprising as it is known that people with spinal pathology often have altered gait patterns, including shorter stride lengths, compared to healthy people [36], suggesting that this gait characteristic may be primarily due to LSS. The Lasso regression and Pearson analysis revealed that PT, LL, sagittal ROM of the hip, and axial ROM of the hip were significantly correlated with stride length. However, the multiple linear regression analysis showed that waist sagittal ROM and LL were not significant mediators. This suggests that other factors, such as muscle strength, balance, and proprioception, may play a more substantial role in determining stride length in these patient populations [37]. These findings are consistent with previous research that has found that stride length is not always significantly different between individuals with and without gait disorders. Higginson et al. [38] found that stride length was not significantly different between individuals with cerebral palsy and typically developing individuals, despite significant differences in other gait parameters. Similarly, a study by Harada et al. [39] found that stride length was not significantly different between older adults with and without a history of falls.

### 4.6. Total Stance Phase

The total stance phase is a critical parameter that reflects the ability of the foot and ankle to provide support and balance during walking. Previous research has shown that lumbar spinal stenosis (LSS) patients have a shorter total stance phase and lower gait velocities compared to healthy controls [40].

Similarly, older adults tend to have slower gait velocities and longer total stance phases than younger adults due to age-related changes in muscle strength and joint mobility [41]. However, in our study, we found no significant difference in the total stance phase between the DLS and LSS groups, suggesting that this gait characteristic may be primarily due to LSS.

Nonetheless, we identified a significant correlation between the pelvic incidence minus sacral slope (PI-SS) and the total stance phase (r = −0.488, *p* = 0.045) and double stance phase (r = −0.451, *p* = 0.041). This finding is consistent with previous research that has demonstrated a relationship between sagittal imbalance and gait parameters [33]. The PI-SS is an essential spinal parameter that reflects the balance between the pelvis and the spine. A high PI-SS indicates that the pelvis is more retroverted than the spine, whereas a low PI-SS indicates that the pelvis is more anteverted than the spine. The correlation between the PI-SS and the total stance phase suggests that spinal balance plays a critical role in the ability of the foot and ankle to provide support and balance during walking. In other words, an imbalance in the sagittal plane can affect the total stance phase, which may ultimately impact gait function in DLS patients.

### 4.7. ROM

Previous studies have shown that sagittal spinal alignment parameters, rather than the coronal Cobb angle, have a greater impact on ROM [19,20]. However, our study found that although the coronal deformity of DLS is usually mild, we still found its impact on activity limitation.

The ROM limitation of the lower extremity in patients with DLS can be attributed to several factors. First, three-dimensional structural changes in the spine, pelvis, and hip may lead to clinically observed stiffness [42,43]. Secondly, as mentioned above, the sagittal imbalance in patients with DLS leads to rotational compensation of the pelvis, resulting in limited movement; thirdly, the reduction in movement can be considered as a compensatory mechanism to limit the progression of the imbalance in the coronal plane of the upper body [28] and finally, the reduction in movement due to the coronal plane may be the result of prolonged bilateral electrical activity (EMG) of the muscles connected to the pelvis. Several studies have shown that dorsal muscle activity is asymmetric on the concave and convex sides of scoliosis, and these changes do not require severe scoliosis to produce the observed differences, even in mild scoliosis and mild radiographic coronal imbalance [44].

### 4.8. Hip Sagittal ROM

Several gait parameters are associated with sagittal hip ROM. Lumbar spinal stenosis (LSS) and spinal deformities such as adolescent idiopathic scoliosis (AIS) can significantly affect hip sagittal ROM, leading to impaired gait patterns and increased risk of falls. A study by Hasegawa et al. found that LSS patients had significantly lower hip flexion and extension ROM compared to healthy controls, and this was associated with lower walking speed and stride length [45]. Similarly, a study by Fritz et al. found that patients with LSS had reduced hip flexion and extension ROM during gait, which was related to lumbar lordosis and pelvic tilt [46]. In our study, we found that hip sagittal ROM was worse in DLS than in LSS, although the difference was not statistically significant, likely because we excluded the effect of LSS. However, we found that lumbar lordosis (LL) was correlated with sagittal ROM of the hip (r = −0.542, *p* = 0.037), suggesting that the alteration of hip sagittal ROM may be related to compensatory mechanisms to maintain balance and compensate for spinal deformities such as DLS.

Moreover, Monticone et al. found that AIS patients had reduced hip flexion and extension ROM compared to healthy controls, suggesting that scoliosis can also affect hip sagittal ROM [40]. Our study also found that the Cobb angle was correlated with sagittal ROM of the hip (r = 0.615, *p* = 0.015), indicating that altered hip sagittal ROM may be related to compensatory mechanisms to maintain balance and compensate for scoliosis [20]. Kawkabani et al. suggested that these dynamic changes made them more likely to fall [26].

### 4.9. Limitations

However, our study has some limitations that should be considered for future research. An increased number of groups, including different severities of DLS and other spinal deformities, could be included to better understand the impact of spinal parameters on gait characteristics. And we did not directly measure quality of life, particularly concerning gait.

Additionally, electromyographic (EMG) studies may be useful to explore muscle activation patterns and their contributions to gait alterations in patients with DLS and scoliosis. By addressing these limitations in future studies, we can better comprehend the complex relationship between spinal parameters, ROM, and gait characteristics, ultimately improving treatment and management strategies for patients with spinal deformities.

## 5. Conclusions

This study illuminates significant differences in gait characteristics and hip mobility between patients with degenerative lumbar scoliosis (DLS) and those with lumbar spinal stenosis (LSS), thereby highlighting the unique challenges associated with managing DLS. Importantly, our findings pave the way for personalized treatment strategies. We identified a Cobb angle cut-off of 16.1 as a critical threshold for predicting gait abnormalities, particularly for walking speeds below 0.8 m/s. Patients exceeding this threshold may require targeted interventions, including focused attention on lower limb conditions, to improve their quality of life.

Furthermore, our study elucidates the specific sagittal and coronal spinal parameters, such as Cobb angle, T1PA, CSVA, PI, PT, LL, and PI-SS, that significantly influence gait in DLS patients. These correlations not only offer insights into the underlying biomechanical and neurological mechanisms affecting gait but also provide clinicians with actionable data for surgical planning

Furthermore, our study elucidates the specific sagittal and coronal spinal parameters, such as Cobb angle, T1PA, CSVA, PI, PT, LL, and PI-SS, that significantly influence gait in DLS patients. These correlations not only offer insights into the underlying biomechanical and neurological mechanisms affecting gait but also provide clinicians with actionable data for surgical planning.

These findings underscore the need for comprehensive assessments and personalized interventions in managing spinal disorders. Future research should expand the study population and include electromyographic assessments to deepen our understanding of the underlying mechanisms, thereby facilitating more effective treatment strategies.

## Figures and Tables

**Figure 1 jpm-13-01576-f001:**
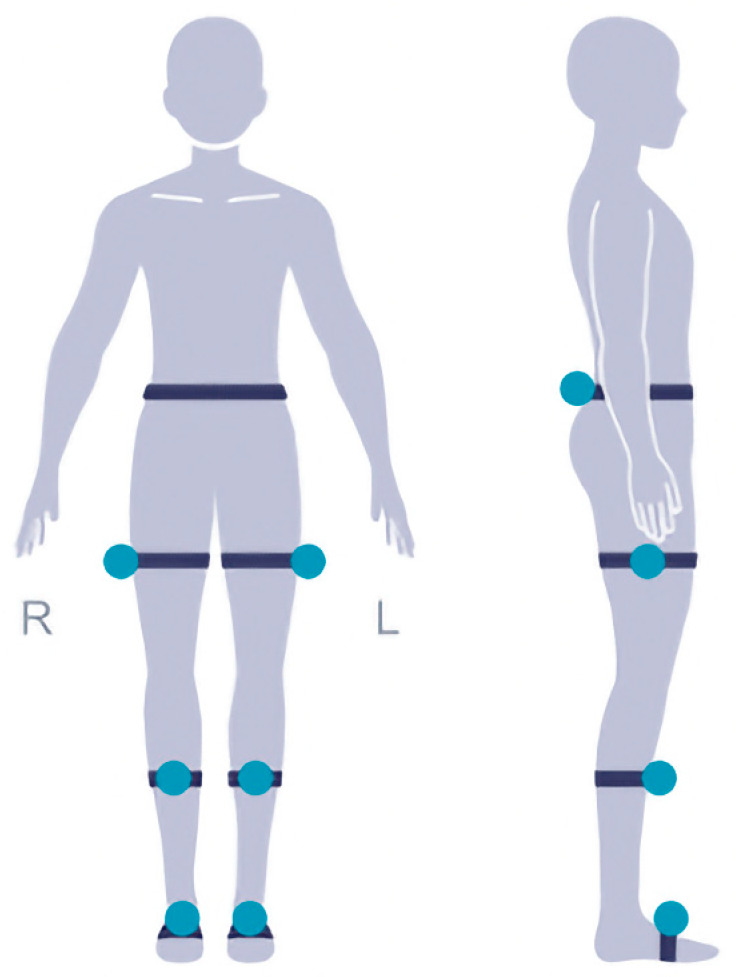
Placement of sensors for gait analysis.

**Figure 2 jpm-13-01576-f002:**
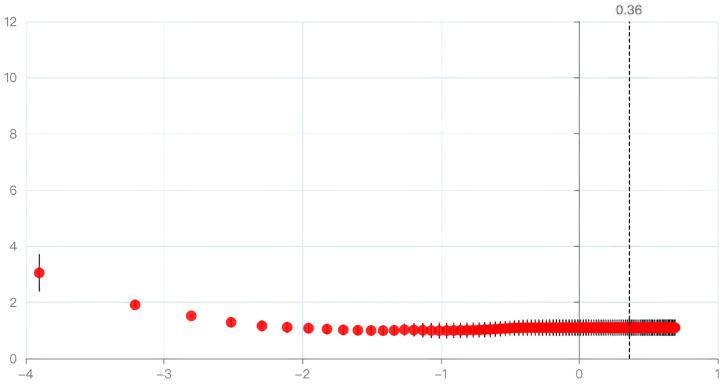
Cross-validation for λ selection. Red dots represent individual predictors’ coefficients. The vertical dashed line at λ = 0.36 indicates the optimal value, beyond which most coefficients approach zero.

**Figure 3 jpm-13-01576-f003:**
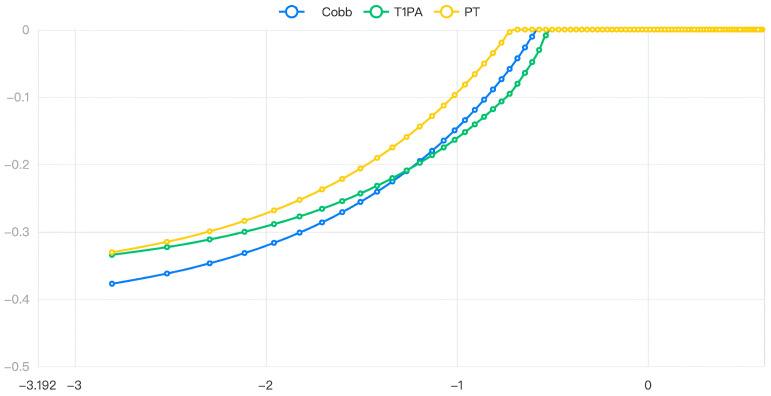
Relationship between λ and model regression coefficients.

**Table 1 jpm-13-01576-t001:** Demographics between DLS group and LSS group.

Statistics	DLS Group	LSS Group	*p*
Gender			0.78
Male	4	8	
Female	16	22	
Age (y)	67 ± 4.59	63 ± 5.53	0.12
Height (cm)	158.33 ± 6.21	163.41 ± 7.41	0.07
Weight (kg)	64.66 ± 12.53	73.08 ± 11.22	0.08
BMI (kg/m^2^)	25.69 ± 3.92	27.45 ± 4.41	0.28

DLS: degenerative lumbar scoliosis; LSS: lumbar spinal stenosis; BMI: body mass index.

**Table 2 jpm-13-01576-t002:** The spinopelvic parameters between DLS group and LSS group.

	DLS Group	Control Group	*p*
CSVA	27.72 ± 14.68	3.51 ± 5.58	0.001 *
T1PA	32.29 ± 9.26	16.55 ± 7.61	0.002 *
LL	21.76 ± 15.09	44 ± 24.02	0.021 *
PT	28.26 ± 8.36	15.8 ± 6.18	0.004 *
SS	25.7 ± 9.28	40.91 ± 14.84	0.012 *
PI-LL	32.2 ± 14.77	12.71 ± 11.13	0.010 *
Cobb	24.02 ± 13.17	2.53 ± 6.20	0.05 *

DLS: degenerative lumbar scoliosis; LSS: lumbar spinal stenosis; T1PA: T1 pelvic angle; LL: lumbar lordosis; PT: pelvic tilt; SS: sacral slope; *: *p*-value < 0.05.

**Table 3 jpm-13-01576-t003:** Kinematic gait parameters between DLS group and control group.

	DLS Group	Control Group	*p*
Hip sagittal ROM	26.17 ± 6.17	32.83 ± 4.39	0.096
Hip coronal ROM	16.32 ± 5.77	20.48 ± 6.75	0.097
Hip axial ROM	14.2 ± 9.30	15.07 ± 4.14	0.981
Knee sagittal ROM	46.91 ± 9.35	55.91 ± 5.14	0.006 *
Ankle sagittal ROM	25.71 ± 8.20	30.28 ± 7.26	0.143

ROM: range of motion; *: *p*-value < 0.05.

**Table 4 jpm-13-01576-t004:** Kinematic gait parameters between DLS group and control group.

	DLS Group	Control Group	*p*
velocity	0.65 ± 0.15 m/s	0.77 ± 0.23 m/s	0.047 *
cadence	87.05 ± 2.61 apm	102.95 ± 13.98 apm	0.031 *
stride length	0.70 ± 0.26 m	0.85 ± 0.29 m	0.171
total stance phase	59.25%	58.30%	0.807
total swing phase	40.75%	41.70%	0.845
double stance phase	14.22%	14.85%	0.591
gait variability	9.53%	3.45%	0.001 *
gait symmetry	8%	4%	0.040 *

DLS: degenerative lumbar scoliosis; *: *p*-value < 0.05.

**Table 5 jpm-13-01576-t005:** ROC curve analysis for coronal Cobb angle in predicting gait impairment.

Parameter	AUC	Optimal Threshold	Sensitivity	Specificity	Cut-Off Value	Parameter
Coronal Cobb Angle	0.751	0.492	0.692	0.800	16.100	

AUC: area under curve, optimal threshold: best ROC point, cut-off: Cobb angle threshold for gait impairment.

## Data Availability

The data that support the findings of this study are available from our hospital but restrictions apply to the availability of these data, which were used under license for the current study, and so are not publicly available. The data are, however, available from the authors upon reasonable request and with permission of our hospital. Interested parties may contact the first author for further information.

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
