# Peer review of "Lasso Analysis of Gait Characteristics and Correlation with Spinopelvic Parameters in Patients with Degenerative Lumbar Scoliosis"

_jpm, 2023, doi:10.3390/jpm13111576_

Round 1

Reviewer 1 Report

Comments and Suggestions for Authors

Medical practice has undergone tremendous advances(1). and these advances has been widened the scope of field of spinal surgical practice.  In this paper, the authors evaluated the gait characteristics of patients with degenerative lum- 11 bar scoliosis (DLS) and patients with simple lumbar spinal stenosis (LSS) by means of a three-di mensional gait analysis system, aiming to determine the image of spinal deformity on gait and the  correlation between spinal-pelvic parameters and gait characteristics in patients with DLS to assist clinical work.

A total of 50 subjects were enrolled in this study, of 16 whom 20 patients with DLS served as the case group and 30 middle-aged and elderly patients with 17 LSS were selected as the control group

In conclusion, DLS patients demonstrate distinctive gait abnormalities and reduced hip mobility com- pared to LSS patients. Significant correlations between crucial spinopelvic parameters and these  gait changes underline their potential influence on gait disturbances in DLS.

I have some concern about the sudy

The authırs suggest that that their result  can  guide personalized treatment of this patient.

How does this study be usefull in the personalized treatment of patients?

How does this improve the quality of life for DLS patients?

.

What does this study add a new data to current literature, it can be discussed .

References

1.        Kanat A, Tsianaka E, Gasenzer ER, Drosos E. Some Interesting Points of Competition of X-Ray using during the Greco-Ottoman War in 1897 and Development of Neurosurgical Radiology: A Reminiscence. Turk Neurosurg. 2022;32(5):877–81.

Author Response

Thank you for your thoughtful questions and the opportunity to clarify the contributions of our study.

  1. How does this study be useful in the personalized treatment of patients? Our study identifies specific spinal-pelvic parameters that significantly affect gait in patients with degenerative lumbar scoliosis (DLS). For instance, we found that a Cobb angle greater than 16.1 has a notable impact on walking speed. This allows clinicians to focus on these parameters when planning surgical interventions, thereby tailoring the treatment to individual patient needs. By understanding the factors that contribute to reduced range of motion and walking speed, clinicians can make more informed decisions about surgical adjustments, leading to more personalized treatment strategies. For example, understanding that a larger Cobb angle, T1PA, and higher CSVA tend to result in slower walking speeds, and that larger PI, PT, and LL usually correlate with smaller stride lengths, allows for more targeted surgical adjustments.

  2. How does this improve the quality of life for DLS patients? Thank you for your insightful comments. You are correct; our study does not directly measure the impact on the quality of life for DLS patients through clinical questionnaires. However, we aim to address a frequent issue reported by DLS patients, which is difficulty in walking. By identifying key spinal-pelvic parameters that influence gait, we provide a basis for targeted surgical and non-surgical interventions that could potentially alleviate these walking issues. We believe that improving these gait abnormalities can, in turn, enhance the quality of life for DLS patients. We acknowledge this limitation and have included it in the "Limitations" section of our paper for future research. Thank you for bringing this to our attention.

  3. What does this study add to new data to current literature? Our study fills a gap in the literature by focusing on the gait characteristics of DLS patients, a subject that has been relatively underexplored. Additionally, we employ Lasso regression, a machine learning technique, to handle the complex, multi-dimensional data generated by gait analysis systems. This innovative approach allows us to sift through large datasets effectively and identify the most critical parameters affecting gait. Thus, our study not only adds new data but also introduces a novel methodological approach to the field.

We appreciate your feedback and will make the necessary revisions to our manuscript to address these points more explicitly.

Reviewer 2 Report

Comments and Suggestions for Authors

The author have done a Lasso analysis of gait characteristics and correlated with spino-pelvic parameters in patients with DLS. The author have compared between 20 patients with DLS and 30 patients with LSS.

Overall comments :

1. It would have been better to compare with normal subjects. Patients with LSS also will have alteration in gait depending on the severity of stenosis and altered conditions.

2. The author should have commented on change in spino- pelvic parameters between control and study group in the abstract.

3. Changes should have compared between aligned and malaligned groups.

4. Please clarify as to how the knowledge of gait alteration will guide personalised treatment and intervention strategies.

Specific comments :

1. Introduction:

                  Should be very specific to gait alterations in DLS - what is known, what is lacunae in literature and need for study.

This aspect of change of gait in DLS has been earlier studied.

2. Materials and methods:

                  Well written.

3. Discussion :

                  Adequate , comprehensive.

Author Response

Thank you for your thoughtful comments. Here are our responses to your concerns:

1、We initially considered comparing our DLS patients with a normal control group. However, given the specific nature of degenerative scoliosis in China, where most patients seek medical attention due to back and leg pain, we decided that using LSS patients as a control group would be more appropriate. This approach allows us to isolate the impact of DLS on gait abnormalities from the long-term effects of back and leg pain.

2、We appreciate your suggestion to comment on the changes in spinal-pelvic parameters between the control and study groups in the abstract. We have discussed the possible reasons for these changes in the discussion section and have now updated the abstract to reflect this. Our study primarily aims to identify the factors affecting gait through advanced statistical methods like machine learning, which can guide surgical adjustments tailored to individual patient symptoms.

3、Thank you for your insightful suggestion. We have re-analyzed our data based on CSVA greater than 5 degrees and an absolute value of PI-LL greater than 10 degrees. The results were largely consistent with our previous findings based on Cobb angle groupings. This could be due to the more severe imbalance conditions in the DLS patient group. While this new analysis has not been directly incorporated into the original manuscript, we are open to discussing its inclusion in an appropriate manner if deemed necessary.

4、Our study is useful for personalized treatment in several ways. First, we found that a Cobb angle greater than 16.1 significantly impacts walking speed. Therefore, patients exceeding this threshold should be closely monitored for lower limb conditions. Second, we identified specific spinal-pelvic parameters that influence gait abnormalities, such as reduced range of motion in the lower limbs and decreased walking speed. These findings allow for targeted surgical adjustments to correct these issues, offering a more personalized treatment approach.

5、

“Introduction:Should be very specific to gait alterations in DLS - what is known, what is lacunae in literature and need for study.This aspect of change of gait in DLS has been earlier studied”:While previous studies have indeed explored gait alterations in DLS, our study brings a novel approach to the table. Traditional statistical methods often fall short in handling the large and diverse set of data generated by gait analysis instruments. Our study employs advanced statistical methods like Lasso and machine learning to better analyze this data, offering a more nuanced understanding of the factors affecting gait in DLS patients. We will make sure to highlight this innovative aspect in the introduction section of our manuscript.

Reviewer 3 Report

Comments and Suggestions for Authors

The study could be improved by changing the design. I suggest doing the study before and after a surgical correction of the scoliosis and analyzing the differences. Each patient would the his/her own control.

Author Response

1. Regarding the quality of English, we appreciate your honesty. We will ensure that the manuscript undergoes rigorous proofreading by native English speakers to improve its language quality.

2. Your suggestion to conduct a pre-and-post surgical correction study is indeed valuable. While the current study aims to provide a foundational understanding of how spinal-pelvic parameters affect gait in DLS patients, your idea offers a compelling avenue for future research. Implementing a before-and-after surgical design would indeed provide each patient as his/her own control, thereby eliminating many confounding variables. We will consider incorporating this design in our future studies to further validate and extend the findings of the current study.

Thank you for your constructive feedback. We will make the necessary revisions to improve the quality of the manuscript.